# The impact of believing you have had COVID-19 on self-reported behaviour: Cross-sectional survey

**Louise E. Smith**[1,2]*, **Abigail L. Mottershaw**[3], **Mark Egan**[3], **Jo Waller**[4], **Theresa M. Marteau**[5], **G. James Rubin**[1,2]

**1** King's College London, Institute of Psychiatry, Psychology and Neuroscience, London, United Kingdom, **2** NIHR Health Protection Research Unit in Emergency Preparedness and Response, London, United Kingdom, **3** Behavioural Insights Team, London, United Kingdom, **4** Faculty of Life Sciences and Medicine, King's College London, London, United Kingdom, **5** Behaviour and Health Research Unit, Institute of Public Health, University of Cambridge, Cambridge, United Kingdom

* louise.e.smith@kcl.ac.uk

**Data Availability Statement:** We have uploaded the dataset to the Open Science Framework. The DOI for the dataset is: 10.17605/OSF.IO/Y7EHA.

## Abstract

### Objectives

To investigate whether people who think they have had COVID-19 are less likely to report engaging with lockdown measures compared with those who think they have not had COVID-19.

### Design

On-line cross-sectional survey.

### Setting

Data were collected between 20th and 22nd April 2020.

### Participants

6149 participants living in the UK aged 18 years or over.

### Main outcome measures

Perceived immunity to COVID-19, self-reported adherence to social distancing measures (going out for essential shopping, nonessential shopping, and meeting up with friends/family; total out-of-home activity), worry about COVID-19 and perceived risk of COVID-19 to oneself and people in the UK. Knowledge that cough and high temperature / fever are the main symptoms of COVID-19. We used logistic regression analyses and one-way ANOVAs to investigate associations between believing you had had COVID-19 and binary and continuous outcomes respectively.

### Results

In this sample, 1493 people (24.3%) thought they had had COVID-19 but only 245 (4.0%) reported having received a positive test result. Reported test results were often incongruent

The link for the study page, where the dataset is stored is https://osf.io/y7eha/.

**Funding:** JW is funded by a career development fellowship from Cancer Research UK (ref C7492/A17219). LS and GJR are supported by the National Institute for Health Research Health Protection Research Unit (NIHR HPRU) in Emergency Preparedness and Response, a partnership between Public Health England, King's College London and the University of East Anglia. The views expressed are those of the authors and not necessarily those of the NIHR, Public Health England or the Department of Health and Social Care. Data collection was funded via a block Government grant to the Behavioural Insights Team. The funders had no role in study design, data collection and analysis, decision to publish, or preparation of the manuscript.

**Competing interests:** JW is funded by a career development fellowship from Cancer Research UK (ref C7492/A17219). Data collection was funded via a block Government grant to the Behavioural Insights Team. There are no patents, products in development or marketed products associated with this research to declare.

with participants' belief that they had had COVID-19. People who believed that they had had COVID-19 were: more likely to agree that they had some immunity to COVID-19; less likely to report adhering to lockdown measures; less worried about COVID-19; and less likely to know that cough and high temperature / fever are two of the most common symptoms of COVID-19.

## Conclusions

At the time of data collection, the percentage of people in the UK who thought they had already had COVID-19 was about twice the estimated infection rate. Those who believed they had had COVID-19 were more likely to report leaving home. This may contribute to transmission of the virus. Clear communications to this growing group are needed to explain why protective measures continue to be important and to encourage sustained adherence.

## Introduction

Since the onset of the COVID-19 outbreak, numerous countries have introduced "lockdown" measures to limit contact between people and reduce the spread of illness. In the UK, these measures were introduced on 23[rd] March 2020 [1], permitting outings only for very limited reasons (to go shopping for food and other essentials such as medicine, for exercise, and to help or provide care for a vulnerable person). Visiting friends or family in other households was not permitted, and neither was shopping for nonessential items. People also had to work from home wherever possible. Rules were eased slightly on the 11[th] May [2], and again on 4[th] July 2020 [3]. Countries around the world are pinning their hopes on testing as part of their exit strategy from more severe measures. This includes antigen testing for current COVID-19 infection, and antibody testing, which tests if people have had COVID-19 in the past. However, the World Health Organization has warned against using antibody tests in order to issue people with "immunity passports" due to fears that those who test positive for antibodies may stop adhering to protective measures [4]. The UK Government has named antigen testing and contact tracing as one of the key pillars of the UK recovery strategy [2], and the NHS Test and Trace system was launched on 28[th] May 2020 in England [5].

In light of the emphasis on antigen testing in the UK's COVID-19 recovery strategy, it is especially important to know whether people who believe they have had COVID-19 are less likely to adhere to protective measures. For example, people who believe they have had COVID-19 may be more likely to think they are completely immune, stop engaging in protective behaviours such as handwashing and reduce distancing themselves physically from others. This may contribute to transmission of the virus for two reasons. First, test results can be wrong [6] and in the absence of testing people can misdiagnose themselves: this can lead people to believe that they have had COVID-19 when they have not. Second, for people who have had COVID-19, it remains unknown whether they could catch COVID-19, and be infectious, more than once [4]. Evidence suggests that COVID-19 antibody levels may decrease over time after contracting COVID-19 [7]. Therefore, it remains likely that people will be required to adhere to protective measures for COVID-19 even if they have had the illness previously. However, there is currently no evidence about whether adherence to protective measures differs based on belief that you have had COVID-19 (either self-diagnosed or confirmed by an antigen or antibody test).

In this study, we explored whether believing that you have already had COVID-19 was associated with self-reported behaviour early in the pandemic. We hypothesised that people who think they have had COVID-19 are: more likely to believe that they are immune to COVID-19; less likely to adhere to social distancing measures; less worried about COVID-19; and perceive a lower risk of COVID-19 to themselves, but no difference in perceived risk of COVID-19 to others. We also investigated awareness of the most common COVID-19 symptoms as a marker of likely accuracy of self-diagnosis.

## Method

### Design

This cross-sectional survey was carried out by the Behavioural Insights Team on their in-house online experimentation platform, Predictiv, between 20th and 22nd April.

### Participants

Participants (n = 6149) were recruited from Predictiv's research panel (n = 500,000 UK adults) and were eligible for the study if they were aged 18 or over and lived in the UK. The use of online research panels allows for quick data collection from a large number of participants. Quota sampling, fulfilling pre-specified targets based on age, gender, income and region, was used to ensure the sample that was broadly representative of the general UK population [8]. This method of sampling allows proportionate representation of target groups in the sample that may otherwise be under-represented (e.g. older adults). 89% of people who clicked on the link subsequently completed the study materials. For this survey, participants were reimbursed in points (equivalent to up to approximately £1) which could be redeemed in cash, gift vouchers or charitable donations. Participants did not know the topic of the survey before commencing it.

### Study materials

These questions were asked as part of an experimental study investigating self-reported behavioural outcomes of antibody test terminology [9]. Results from participants included in this study, reporting on the experimental study have been published in a separate peer-reviewed publication [10]. There is no overlap in results reported in this current study and the experimental study. For the purposes of this paper, we collapsed the data across all arms of the experiment and controlled for experimental condition.

Survey materials are presented in the supporting information (S1 File).

**Having had COVID-19.** Widespread antigen testing was not available at the time of data collection; only frontline essential workers who had symptoms were eligible to request an antigen test [11]. Therefore we asked participants if they thought they had "already had coronavirus?" Response options were "Yes, definitely", "Yes, probably", "No, probably not", and "No, definitely not".

**Other measures.** We asked participants if they had been tested for COVID-19. Possible answers included "yes, the results showed I did have coronavirus", "yes, the results showed I did not have coronavirus" and "no, I haven't been tested".

To measure perceived immunity to COVID-19, we asked participants to what extent they agreed or disagreed with the statement "I think I have some immunity to coronavirus" on a five-point Likert scale ("strongly agree" to "strongly disagree").

We asked participants to state "over the last seven days, on how many days" they had: been to the shops, for groceries/pharmacy, been to the shops, for things other than groceries/

pharmacy, gone for a walk or some other exercise; gone out to work, helped or provided care for a vulnerable person, and met up with friends and/or family they did not live with. At the time of data collection, the only outings allowed were to go to the shops for groceries/pharmacy, to go for a walk or some other exercise, and to help or provide care for a vulnerable person; people could only go to work if they could not work from home [1].

We asked participants to rate how worried they were about COVID-19 on a five-point Likert-type scale from "not at all worried" to "extremely worried". We also asked participants to rate the extent to which they thought COVID-19 posed a risk to themselves personally and to people in the UK on a four-point Likert-type scale from "no risk at all" to "major risk".

To assess the likelihood of misdiagnosis, we asked participants what they thought "the most common symptoms of coronavirus" were from a list of thirteen items (including cough, high temperature / fever, shortness of breath / difficulty breathing, runny or blocked nose, aches and pains, chest pain, chills / shivering, sore throat, diarrhoea, headaches, stomach ache, feeling tired or having low energy, and loss of sense of smell / taste). Participants could select up to three symptoms.

**Social and demographic characteristics.** Participants were asked to state their: age; gender; employment status; highest educational attainment; and region. Participants were also asked what sector they worked in (to identify key workers) and whether they had children.

## Ethics

Ethical approval for this study was granted by the King's College London Research Ethics Committee (reference: MRA-19/20-18485).

Submission of completed study materials implied consent to take part in the study. Participants were informed of this before starting the study.

## Patient and public involvement

Due to the rapid nature of this research, the public was not involved in the development of the survey materials.

## Power

A sample size of 6,150 allows a 95% confidence interval of plus or minus 1% for the prevalence estimate for each survey item.

## Analysis

**Recoding variables.** We recoded thinking you have had COVID-19 into a binary variable (yes / no), grouping together responses of "Yes, definitely" and "Yes, probably", versus "No, probably not" and "No, definitely not".

We created a binary variable to identify whether participants had correctly identified cough and high temperature / fever as two of the most common symptoms of COVID-19. At the time of data collection, only cough and high temperature / fever were listed as key symptoms of coronavirus; loss or change of sense of smell or taste was added on 18th May 2020 [12]. We coded those who answered "don't know" as incorrect.

We defined non-adherence to social distancing measures by considering the instructions from the UK Government to members of the public that were in force at the time of data collection [13]. If participants went out to the shops for items other than groceries/pharmacy once or more in the last seven days, or met up with friends and/or family they did not live with once or more in the last seven days, we classed them as not adhering to the guidelines. There is

no objective guidance on the frequency of shopping for basic necessities such as food or medi-
cine, with guidance in place at the time of data collection stating that it "must be as infrequent
as possible" [1]. We created a binary variable for shopping for groceries/pharmacy grouping
together those who had been shopping for necessities on two or more days in the last week,
compared to one day or less. We also created a continuous variable representing the total
amount of out-of-home activity a participant had engaged in during the past week, by sum-
ming the number of days they had left the house for each of six activities (shopping for grocer-
ies/pharmacy, shopping for items other than groceries/pharmacy, going for a walk or some
other exercise, going out to work, helping or providing care for a vulnerable person; meeting
up with friends and/or family they did not live with).

**Analyses.** For all analyses with binary outcomes (correct identification of the most com-
mon symptoms of COVID-19; non-adherence to social distancing measures), we used binary
logistic regressions to investigate univariable associations between thinking you have had
COVID-19 and dependent variables. We then used a second logistic regression adjusting for
all social and demographic characteristics (gender, age, presence of a dependent child, employ-
ment status, working in a key sector, highest educational or professional qualification, and
region) and experimental group.

For analyses with a continuous outcome (perceived immunity to COVID-19; worry about
COVID-19; perceived risk of COVID-19 to oneself; perceived risk of COVID-19 to people in
the UK; out-of-home activity), we used a series of one-way ANOVAs to investigate univariable
associations between thinking you have had COVID-19 and dependent variables. We then
used a series of ANCOVAs adjusting for all social and demographic characteristics (gender,
age, presence of a dependent child, employment status, working in a key sector, highest educa-
tional or professional qualification, and region) and experimental group.

Our analyses report unweighted statistics. We corrected for multiple comparisons using a
Bonferroni adjustment ($p$ = .005).

To provide a graphical illustration of the results, we used a bar chart to show the differences
between those who did and did not think they had had COVID-19 in terms of the proportions
giving responses at the extreme end of the scale for relevant outcomes (e.g. strongly agreeing
they have some immunity, being not at all worried about COVID-19).

**Sensitivity analyses.** We re-ran analyses excluding those who had been tested for
COVID-19.

## Results

Only adjusted analyses are reported narratively; unadjusted analyses are reported in the tables.

### Participants

24.3% (n = 1493) of participants thought that they had had COVID-19. Only 9.4% (n = 575)
participants reported having an antigen test for COVID-19. Of those who had been tested,
42.6% (n = 245) reported that the test showed they did have COVID-19, while 57.4% (n = 330)
reported that the test showed they did not have COVID-19. Of those who reported that their
test showed they did not have COVID-19, 56.7% (n = 187) nonetheless thought that they had
had COVID-19. Conversely, of those who reported that the test showed they did have
COVID-19, 22.9% (n = 56) thought that they had not had COVID-19. Personal characteristics
of participants broadly reflect those of the UK general population (Table 1).

Younger participants, those who had a child, those who were employed (full-time, part-
time, or self-employed), and those who worked in a key sector were more likely to report
thinking that they had had COVID-19 (see Table 1).

**Table 1. Associations between participant social and demographic characteristics and thinking you have had COVID-19.**

| Participant characteristics | Level | Had COVID-19 | | Odds ratio (95% CI) | Adjusted odds ratio (95% CI)† |
|---|---|---|---|---|---|
| | | Think have not had COVID-19 n = 4656 n (%) | Think have had COVID-19 n = 1493 n (%) | | |
| Gender | Male | 2197 (75.9) | 697 (24.1) | Reference | Reference |
| | Female | 2459 (75.5) | 796 (24.5) | 1.02 (0.91 to 1.15) | 0.99 (0.87 to 1.12) |
| Age | 18 to 24 years | 1003 (70.5) | 419 (29.5) | Reference | Reference |
| | 25 to 34 years | 823 (67.3) | 400 (32.7) | 1.16 (0.99 to 1.37) | 1.07 (0.89 to 1.28) |
| | 35 to 44 years | 751 (71.9) | 294 (28.1) | 0.94 (0.79 to 1.12) | 0.80 (0.66 to 0.98) |
| | 45 to 54 years | 554 (77.2) | 164 (22.8) | 0.71 (0.58 to 0.87)* | 0.62 (0.49 to 0.78)* |
| | 55 years and over | 1525 (87.6) | 216 (12.4) | 0.34 (0.28 to 0.41)* | 0.36 (0.29 to 0.44)* |
| Have a child | No | 2005 (76.4) | 621 (23.6) | Reference | Reference |
| | Yes | 2386 (75.5) | 776 (24.5) | 1.05 (0.93 to 1.19) | 1.30 (1.14 to 1.50)* |
| Employment status | Not working | 1714 (82.8) | 357 (17.2) | Reference | Reference |
| | Working | 2871 (71.9) | 1124 (28.1) | 1.88 (1.65 to 2.15)* | 1.24 (1.05 to 1.46) |
| Working in key sector | No | 3105 (80.5) | 753 (19.5) | Reference | Reference |
| | Yes | 1551 (67.7) | 740 (32.3) | 1.97 (1.75 to 2.21)* | 1.52 (1.32 to 1.75)* |
| Highest educational or professional qualification | GCSE/vocational/A-level/No formal qualifications | 3382 (76.1) | 1060 (23.9) | Reference | Reference |
| | Degree or higher (Bachelors, Masters, PhD) | 1200 (74.3) | 415 (25.7) | 1.10 (0.97 to 1.26) | 1.10 (0.95 to 1.26) |
| Region | Midlands | 781 (75.7) | 251 (24.3) | Reference | Reference |
| | South & East | 1369 (76.7) | 416 (23.3) | 0.95 (0.79 to 1.13) | 0.98 (0.81 to 1.19) |
| | North | 1120 (77.0) | 335 (23.0) | 0.93 (0.77 to 1.12) | 0.95 (0.78 to 1.17) |
| | London | 701 (70.1) | 299 (29.9) | 1.33 (1.09 to 1.62)* | 1.10 (0.89 to 1.36) |
| | Wales, Scotland and Northern Ireland | 685 (78.1) | 192 (21.9) | 0.87 (0.70 to 1.08) | 0.89 (0.71 to 1.12) |

*p≤.005.

†Adjusting for all other social and demographic characteristics and experimental group.

## Differences between those who think they have and have not had COVID-19

18.5% of participants (n = 1140) agreed or strongly agreed that they had some immunity to COVID-19. Those who thought they had had COVID-19 were more likely to agree that they had some immunity to COVID-19 (did not think they had had COVID-19: 10.7%, n = 500; thought they had had COVID-19: 42.9%, n = 640; Table 2 and Fig 1).

In the last seven days, 38.9% (n = 2389) reported going out to the shops for groceries/pharmacy on two or more days; 29.8% (n = 1833) reported going out to the shops for items other than groceries/pharmacy once or more; and 14.3% (n = 878) reported meeting up with friends and/or family they did not live with once or more. Those who thought they had had COVID-19 were less likely to adhere to social distancing measures and went out shopping for groceries/pharmacy more frequently (Table 3). They also went out more times in total in the last seven days (Table 2).

50.8% (n = 3132) reported being very or extremely worried about COVID-19. Those who thought they had had COVID-19 were less worried about COVID-19 (see Table 2).

17.7% (n = 1091) perceived a major risk of COVID-19 to themselves, while 47.0% (n = 2893) perceived a major risk of COVID-19 to people in the UK. There was no evidence for an association between thinking you had had COVID-19 and perceived risk of COVID-19 (see Table 2).

**Table 2. Associations between thinking you have had COVID-19 and perceived immunity to COVID-19; worry about COVID-19; perceived risk of COVID-19 (to oneself and people in the UK); and total out-of-home activities in the last seven days (continuous outcomes).**

| Participant characteristics | Level | Had COVID-19 | | | | | | | |
|---|---|---|---|---|---|---|---|---|---|
| | | Think have not had COVID-19 n = 4656 | Think have had COVID-19 n = 1493 | Unadjusted analyses | | | Adjusted analyses | | |
| | | | | F | p | $\eta^2$ | F | p | $\eta^2$ |
| I think I have some immunity to COVID-19 | 1 = strongly disagree to 5 = strongly agree | M = 2.38, SD = 1.01 | M = 3.33, SD = 1.01 | 998.11 | < .001* | .14 | 723.59 | < .001* | .11 |
| Total out-of-home activity in the last seven days | Range = 0 to 42 | M = 6.69, SD = 5.63 | M = 9.35, SD = 7.69 | 209.28 | < .001* | .03 | 83.70 | < .001* | .01 |
| Worry about COVID-19 | 1 = not at all worried to 5 = extremely worried | M = 3.59, SD = 1.01 | M = 3.38, SD = 1.12 | 50.16 | < .001* | .01 | 28.52 | < .001* | .01 |
| Perceived risk of COVID-19 to oneself | 1 = no risk at all to 4 = major risk | M = 2.81, SD = 0.76 | M = 2.81, SD = 0.76 | 0.01 | .93 | < .001 | 6.85 | .01 | .001 |
| Perceived risk of COVID-19 to people in the UK | 1 = no risk at all to 4 = major risk | M = 3.39, SD = 0.67 | M = 3.30, SD = 0.70 | 18.04 | < .001* | .003 | 5.67 | .02 | .001 |

*p≤.005.

†Adjusting for all social and demographic characteristics and experimental condition.

59.1% (95% CI 57.8% to 60.3%, n = 3632) correctly identified cough and high temperature / fever as two out of the three most common symptoms of COVID-19. Those who thought they had had COVID-19 were less likely to correctly identify these symptoms (see Table 3).

## Sensitivity analyses

Of those who had not been tested for COVID-19 (n = 5574), 20.0% (95% CI 19.0% to 21.1%) thought they had had COVID-19 (n = 1117).

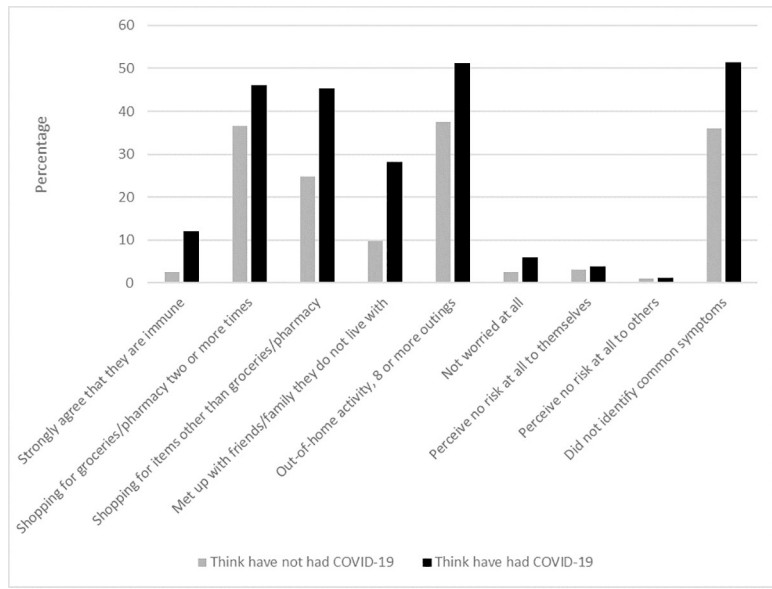

**Fig 1. Psychological and behavioural outcomes for those who did and did not think they had had COVID-19.** Graph depicting differences between people who thought they had had COVID-19 and those who thought they had not had COVID-19 and outcomes (thinking you are immune to COVID-19; shopping behaviour; meeting up with friends/family; out-of-home activity; worry about COVID-19; perceived risk of COVID-19; and ability to identify symptoms of COVID-19).

**Table 3. Associations between thinking you have had COVID-19 and correct identification of most common symptoms of COVID-19; and adherence to social distancing measures (shopping for groceries/pharmacy, shopping for items other than groceries/pharmacy, and meeting up with friends and/or family who do not live with you; binary outcomes).**

| Thinks have had COVID-19? | Self-reported behaviour n (%) | | Odds ratio (95% CI) | Adjusted odds ratio (95% CI)† |
|---|---|---|---|---|
| | **Shopping for groceries/pharmacy** | | | |
| | On one or fewer days in the last week n = 2389 | On two or more days in the last week n = 3760 | | |
| No | 1701 (36.5) | 2955 (63.5) | Reference | Reference |
| Yes | 688 (46.1) | 805 (53.9) | 0.67 (0.60 to 0.76)* | 0.78 (0.69 to 0.89)* |
| | **Shopping for items other than groceries/pharmacy** | | | |
| | Not at all in the last week n = 1833 | On one or more days in the last week n = 4316 | | |
| No | 1156 (24.8) | 3500 (75.2) | Reference | Reference |
| Yes | 677 (45.3) | 816 (54.7) | 0.40 (0.35 to 0.45)* | 0.51 (0.44 to 0.58)* |
| | **Meeting up with friends or family** | | | |
| | Not at all in the last week n = 5271 | On one or more days in the last week n = 878 | | |
| No | 4200 (90.2) | 456 (9.8) | Reference | Reference |
| Yes | 1071 (71.7) | 422 (28.3) | 0.28 (0.24 to 0.32)* | 0.36 (0.30 to 0.43)* |
| | **Correct identification of cough and fever** | | | |
| | Did not correctly identify common symptoms n = 2390 | Correctly identified common symptoms n = 3632 | | |
| No | 1644 (36.0) | 2927 (64.0) | Reference | Reference |
| Yes | 746 (51.4) | 705 (48.6) | 0.53 (0.47 to 0.59)* | 0.61 (0.54 to 0.69)* |

*p≤.005.

†Adjusting for all social and demographic characteristics and experimental condition.

In adjusted analyses, women were more likely to think that they had had COVID-19 (aOR = 1.16, 95% CI 1.01 to 1.34). There was no evidence for an association between having a child or employment status and thinking you had had COVID-19.

There was also no evidence for an association between thinking you had had COVID-19 and: going shopping for groceries/pharmacy on two or more days in the last week, correct identification of two of the most common symptoms of COVID-19, and total out-of-home activity.

## Discussion

Almost one quarter of participants thought they had had COVID-19. This percentage is higher to that seen in other surveys from the UK, with findings from daily tracker surveys conducted at the time indicating that approximately 10% to 18% thought that they had had COVID-19 [14, 15]. Differences in findings may be explained by the fact that these data only cover dates until 20th April. Although we cannot be sure of the true proportion of the population that had had COVID-19 at the time of data collection, it was likely much lower, with data from the Office for National Statistics (ONS) indicating that approximately 7% of people tested positive for antibodies to COVID-19 between 26th April and 24 May 2020 [16]. It is likely that a substantial element of self-misdiagnosis underlies the high rate that we observed. This is

supported by the high number of participants who felt they had had COVID-19 and who were unable to identify cough and fever as key symptoms of the illness (52.8%). While absolute rates should be taken with caution due to the use of self-reported survey data and possible influence of social desirability bias, associations within the data provide useful insights [17]. In the absence of properly conducted observational studies, our results suggest that adherence to lockdown measures for COVID-19 is lower in people who believe they have previously had the virus. The proportion of the population who believe, rightly or wrongly, that they have had COVID-19 will only increase over time. Understanding how this affects behaviour is therefore important.

A high percentage of people who reported having had an antigen test held beliefs about whether they had had COVID-19 that were incongruous with their test result. Lack of clarity about estimates of sensitivity and specificity for antigen and antibody testing [18] may underlie these findings. For those people who reported having tested negative, we cannot tell if they thought they had developed COVID-19 since receiving this test result. Lack of confidence in antigen test results may have important implications for the NHS Test and Trace system, in which people who test positive for COVID-19 are required to self-isolate for ten days (and any household members for fourteen days) and asked to pass on contact details of their close contacts. Evidence suggests that adherence to self-isolation after developing symptoms of COVID-19 is low [19]. If people do not have confidence in the results of their antigen tests, this may drive adherence down further.

We found that people who thought they had had COVID-19 were more likely to think that they had some immunity to the virus and were less likely to adhere to social distancing measures. In particular, people were less likely to report adhering to measures that were not allowed at all in the UK at the time, such as meeting up with friends and/or family that you did not live with and shopping for nonessentials. They also reported more outings in the last week than those who did not think they had had COVID-19, however this result should be taken with caution as there was no longer an association in our sensitivity analyses. While increased out-of-home activity might be partially explained by social and demographic characteristics, such as age, we adjusted for all social and demographic characteristics in analyses. Given the cross-sectional nature of our data, it is impossible to be clear on causality–it may be that not adhering to social distancing rules leads to a greater likelihood of contracting COVID-19. However, the findings do fit with concerns expressed by the WHO that believing oneself to have had COVID-19 results in reduced adherence to protective behaviours [4].

This finding has important implications at an individual level and at a policy level. Lower perceived social norms were associated with non-adherence to lockdown measures in the UK [19]. There is the potential for a vicious cycle here, with people who believe they have had COVID-19 being less likely to adhere to protective measures [7], lowering social norms and further decreasing adherence [20]. To date, there are no communications specifically targeting those who think they have had COVID-19. This will become increasingly important in minimising transmission as the outbreak continues. Communications should acknowledge the growing proportion of the population who think that they have had COVID-19 and should issue targeted recommendations for this group explaining why it remains important to adhere to personal protective measures put in place to prevent the spread of COVID-19.

In addition to associations with self-reported behaviour, thinking that you had had COVID-19 was associated with decreased worry about COVID-19. This appears logical, however there was little evidence for an association between perceived risk (to oneself and others) and believing you have had COVID-19. This is contrary to evidence finding that those who have higher risk perceptions are more likely to take protective action (in this case staying at home) [21]. However, results may be reflective of uncertainties, especially in the early stages of

the pandemic, surrounding whether it is possible to contract COVID-19 more than once [4], duration of antibody presence after having had COVID-19, and if antibody count varies with age. As we did not measure different factors that may contribute to worry (e.g. concern about personal finances / job, impact on physical / mental health), we are unable to tell which specific worries may be driving this decrease. It should be noted that differences detected in worry about COVID-19, and perceived risk of COVID-19 between those who did and did not think they had had the virus were small and may not be meaningful in real world situations.

Older participants were less likely to think they had had coronavirus. This may be because of a greater proportion of this group who were "shielding" (not leaving the home at all for at least 12 weeks). Those who had a child were more likely to report having had coronavirus, perhaps linked to greater exposure, or perceived exposure, among this group [22, 23]. However, schools in the UK closed on 23rd March 2020 except for children of key workers [24] reducing contacts between children [25]. Results pertaining to having children should be taken with caution as there was no longer any evidence for an association when analysing only those who had not been tested for COVID-19. Those who were employed (full-time, part-time or self-employed) were also more likely to think that they had had COVID-19, as were those working in key sectors. This may be due to increased objective exposure and perceived exposure as these groups continued to go out to work, while others worked from home or stopped working. ONS data indicated that incidence of COVID-19 was higher in patient-facing healthcare workers and resident-facing social care workers compared to others in the UK [26, 27]. There was no longer any evidence for an association between employment and thinking you had had COVID-19 when removing those who had been tested, therefore this interpretation should be taken with caution.

This study has several limitations. First, while quotas were used to ensure a sample that was broadly representative of the general UK population, we cannot be certain whether respondents in survey panels are representative of the general population [28, 29]. Despite this, there are small differences across most topics between online survey respondents and survey respondents that cover the entire public, using mail or telephone surveys [30]. We also cannot rule out participation bias. Given potential participants were not aware of the topic of the survey before starting it, the risk of this was low. Quota samples aim to minimise response bias by filling pre-determined targets so that the social and demographic characteristics of the participants are representative of the national population. As such, participants that belong to a quota that has already been met are prevented from completing the survey. Therefore, response rate is not a useful indicator of response bias in quota samples. Second, we relied on self-reported measures. Due to the possible influence of social desirability bias, it may be that reported rates of adherence to lockdown measures are over-estimates. This presents a worrying picture. In the absence of a properly conducted observational study, results from self-reported data provide useful insights into possible patterns of behaviour change in those who believe they have had COVID-19. Third, we did not differentiate between outings that were in line with Government guidelines and those that were not in our measure of "total out-of-home activity". Third, because we used a cross-sectional study design, we are unable to determine the direction of associations. Fourth, due to the large sample size, small differences between groups were statistically significant. Where detected differences were very small, there may not be a meaningful influence of these differences (e.g. perceived risk to self).

## Conclusions

There is evidence that people who believe they have had COVID-19 are less likely to adhere to protective behaviours put in place to prevent the spread of the virus, such as physical

distancing. Those who thought they had had COVID-19 were also more likely to believe that they had some immunity to the virus. Even when tested, the reported result of an antigen test was not necessarily reflected in people's belief about whether they had had COVID-19. Clear, targeted communications should be used to advise this constantly growing group that protective measures for COVID-19 continue to be important to promote adherence.

## Supporting information

**S1 File. Survey items.**
(DOCX)

## Author Contributions

**Conceptualization:** Louise E. Smith, Jo Waller, Theresa M. Marteau, G. James Rubin.

**Formal analysis:** Louise E. Smith.

**Methodology:** Jo Waller, Theresa M. Marteau, G. James Rubin.

**Project administration:** Abigail L. Mottershaw, Mark Egan.

**Writing – original draft:** Louise E. Smith.

**Writing – review & editing:** Louise E. Smith, Abigail L. Mottershaw, Jo Waller, Theresa M. Marteau, G. James Rubin.

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
