## [Decision Letter · Decision Letter 0]

10 Sep 2020

PONE-D-20-23741

The impact of believing you have had COVID-19 on self-reported behaviour: Cross-sectional survey

PLOS ONE

Dear Dr. Smith,

Thank you for submitting your manuscript to PLOS ONE. After careful consideration, we feel that it has merit but does not fully meet PLOS ONE’s publication criteria as it currently stands. Therefore, we invite you to submit a revised version of the manuscript that addresses the points raised during the review process.

We look forward to receiving your revised manuscript.

Kind regards,

Ramesh Kumar, PhD

Academic Editor

PLOS ONE

Journal Requirements:

2. Please provide additional details regarding participant consent.

In the ethics statement in the Methods and online submission information, please ensure that you have specified (i) whether consent was informed and (ii) what type you obtained (for instance, written or verbal, and if verbal, how it was documented and witnessed).

If your study included minors, state whether you obtained consent from parents or guardians.

If the need for consent was waived by the ethics committee, please include this information.

3. In your Supporting Information section, please provide a copy of your questionaire used in this study if available.

'JW is funded by a career development fellowship from Cancer Research UK (ref C7492/A17219). LS and GJR are supported by the National Institute for Health Research Health Protection Research Unit (NIHR HPRU) in Emergency Preparedness and Response at King’s College London in partnership with Public Health England (PHE), in collaboration with the University of East Anglia and Newcastle University. The views expressed are those of the authors and not necessarily those of the NHS, the NIHR or the Department of Health and Social Care, Public Health England. Data collection was funded via a block Government grant to the Behavioural Insights Team.'

a. Please complete your Competing Interests statement to state any Competing Interests. If you have no competing interests, please state "The authors have declared that no competing interests exist.", as detailed online in our guide for authors at http://journals.plos.org/plosone/s/submit-now

6. We noted in your submission details that a portion of your manuscript may have been presented or published elsewhere:

'Results from analyses of the experimental study investigating self-reported behavioural outcomes of antibody test terminology have been published elsewhere (Waller J, Rubin GJ, Potts HWW, Mottershaw AL, Marteau TM. "Immunity Passports" for SARS-CoV-2: an online experimental study of the impact of antibody test terminology on perceived risk and behaviour. BMJ Open. In press. doi: 10.1101/2020.05.06.20093401); this has been highlighted in the manuscript. '

Please clarify whether this publication was peer-reviewed and formally published.

If this work was previously peer-reviewed and published, in the cover letter please provide the reason that this work does not constitute dual publication and should be included in the current manuscript.

7. Please upload a copy of Figure 1, to which you refer in your text on page 14. If the figure is no longer to be included as part of the submission please remove all reference to it within the text.

Reviewers' comments:

Reviewer's Responses to Questions

**Comments to the Author**

1. Is the manuscript technically sound, and do the data support the conclusions?

Reviewer #1: Yes

Reviewer #2: Yes

2. Has the statistical analysis been performed appropriately and rigorously? 

Reviewer #1: Yes

Reviewer #2: Yes

3. Have the authors made all data underlying the findings in their manuscript fully available?

Reviewer #1: Yes

Reviewer #2: Yes

4. Is the manuscript presented in an intelligible fashion and written in standard English?

Reviewer #1: Yes

Reviewer #2: Yes

5. Review Comments to the Author

Reviewer #1: The author must address the following points before the final acceptance of article.

1. Why author selected 18 years old or above age participants?

2. Must add the name of used statistical tool in the abstract section.

Reviewer #2: The study “The impact of believing you have had COVID-19 on self-reported behaviour: Cross-sectional survey” is interesting. In this paper the authors investigated whether people who think they have had COVID-19 are less likely to report engaging with lockdown measures compared with those who think they have not had COVID-19. The paper is well set, and the contents are clearly described. The authors almost achieved their objectives. However, the following suggestions should be incorporated before resubmitting the paper.

1. A “conclusion” section must be added which concludes the whole paper.

2. The results from Table 1 and Table 2 are hard to read. Please add some graphs/plots which represent these results in pictorial form. Because a picture is worth more than a thousand words.

6. PLOS authors have the option to publish the peer review history of their article (what does this mean?). If published, this will include your full peer review and any attached files.

Reviewer #1: No

Reviewer #2: No

---

## [Author Response · Author response to Decision Letter 0]

17 Sep 2020

Journal Requirements:

 The manuscript is now formatted in accordance with PLOS ONE’s style requirements.

2. Please provide additional details regarding participant consent.

In the ethics statement in the Methods and online submission information, please ensure that you have specified (i) whether consent was informed and (ii) what type you obtained (for instance, written or verbal, and if verbal, how it was documented and witnessed).

We have now stated in the manuscript that “submission of completed study materials implied consent to take part in the study. Participants were informed of this before starting the study.”

If your study included minors, state whether you obtained consent from parents or guardians.

The study did not include minors.

If the need for consent was waived by the ethics committee, please include this information.

Ethical approval for this study, including consent method, was granted by the King’s College London Research Ethics Committee (reference: MRA-19/20-18485). 

N/A.

3. In your Supporting Information section, please provide a copy of your questionaire used in this study if available.

We have now included the survey items as supporting information (S1 File).

 Noted; if the manuscript is accepted, we will provide the details relevant to access the data.

'JW is funded by a career development fellowship from Cancer Research UK (ref C7492/A17219). LS and GJR are supported by the National Institute for Health Research Health Protection Research Unit (NIHR HPRU) in Emergency Preparedness and Response at King’s College London in partnership with Public Health England (PHE), in collaboration with the University of East Anglia and Newcastle University. The views expressed are those of the authors and not necessarily those of the NHS, the NIHR or the Department of Health and Social Care, Public Health England. Data collection was funded via a block Government grant to the Behavioural Insights Team.'

a. Please complete your Competing Interests statement to state any Competing Interests. If you have no competing interests, please state "The authors have declared that no competing interests exist.", as detailed online in our guide for authors at http://journals.plos.org/plosone/s/submit-now

The Competing Interest statement should read:

'JW is funded by a career development fellowship from Cancer Research UK (ref C7492/A17219). LS and GJR are supported by the National Institute for Health Research Health Protection Research Unit (NIHR HPRU) in Emergency Preparedness and Response, a partnership between Public Health England, King’s College London and the University of East Anglia. The views expressed are those of the authors and not necessarily those of the NIHR, Public Health England or the Department of Health and Social Care. Data collection was funded via a block Government grant to the Behavioural Insights Team’

Thank you. 

6. We noted in your submission details that a portion of your manuscript may have been presented or published elsewhere:

'Results from analyses of the experimental study investigating self-reported behavioural outcomes of antibody test terminology have been published elsewhere (Waller J, Rubin GJ, Potts HWW, Mottershaw AL, Marteau TM. "Immunity Passports" for SARS-CoV-2: an online experimental study of the impact of antibody test terminology on perceived risk and behaviour. BMJ Open. In press. doi: 10.1101/2020.05.06.20093401); this has been highlighted in the manuscript. '

Please clarify whether this publication was peer-reviewed and formally published.

If this work was previously peer-reviewed and published, in the cover letter please provide the reason that this work does not constitute dual publication and should be included in the current manuscript.

Results from participants included in this study have been published elsewhere (BMJ Open). This publication was peer-reviewed and has been published. Work does not constitute dual publication as it reports on results of an experimental study investigating self-reported behavioural outcomes of antibody test terminology. Those results are not included in the current study, nor were any results from this study presented in the BMJ Open paper. Statements to this effect have been added to the manuscript.

7. Please upload a copy of Figure 1, to which you refer in your text on page 14. If the figure is no longer to be included as part of the submission please remove all reference to it within the text.

We apologise for this omission. Figure 1 has now been uploaded as part of the submission.

Reviewers' comments:

5. Review Comments to the Author

Reviewer #1: The author must address the following points before the final acceptance of article.

1. Why author selected 18 years old or above age participants?

We chose to recruit only those aged 18 years and older as we wanted to investigate associations in an adult population.

2. Must add the name of used statistical tool in the abstract section.

We have now added a sentence to the abstract stating that “we used logistic regression analyses and one-way ANOVAs to investigate associations between believing you had had COVID-19 and binary and continuous outcomes respectively.”

Reviewer #2: The study “The impact of believing you have had COVID-19 on self-reported behaviour: Cross-sectional survey” is interesting. In this paper the authors investigated whether people who think they have had COVID-19 are less likely to report engaging with lockdown measures compared with those who think they have not had COVID-19. The paper is well set, and the contents are clearly described. The authors almost achieved their objectives. However, the following suggestions should be incorporated before resubmitting the paper.

1. A “conclusion” section must be added which concludes the whole paper.

We have now added a conclusion section to the paper.

2. The results from Table 1 and Table 2 are hard to read. Please add some graphs/plots which represent these results in pictorial form. Because a picture is worth more than a thousand words.

We have now included a graph depicting differences between people who thought they had had COVID-19 and those who thought they had not had COVID-19 and outcomes (thinking you are immune; shopping behaviour; meeting up with friends/family; out-of-home activity; worry about COVID-19; perceived risk of COVID-19; and ability to identify symptoms of COVID-19).

---

## [Editor Report · Decision Letter 1]

28 Sep 2020

The impact of believing you have had COVID-19 on self-reported behaviour: Cross-sectional survey

PONE-D-20-23741R1

Dear Dr. Smith,

We’re pleased to inform you that your manuscript has been judged scientifically suitable for publication and will be formally accepted for publication once it meets all outstanding technical requirements.

Kind regards,

Ramesh Kumar, PhD

Academic Editor

PLOS ONE
---

## [Editor Report · Acceptance letter]

20 Oct 2020

PONE-D-20-23741R1 

The impact of believing you have had COVID-19 on self-reported behaviour: Cross-sectional survey 

Dear Dr. Smith:

I'm pleased to inform you that your manuscript has been deemed suitable for publication in PLOS ONE. Congratulations! Your manuscript is now with our production department. 

Kind regards, 

on behalf of

Dr. Ramesh Kumar 

Academic Editor

PLOS ONE